# Dynamic Offloading Method for Mobile Edge Computing of Internet of Vehicles Based on Multi-Vehicle Users and Multi-MEC Servers

**Xiaochao Dang [1,2,\*], Lin Su [1], Zhanjun Hao [1,2] and Xu Shang [1]**

[1] College of Computer Science and Engineering, Northwest Normal University, Lanzhou 730070, China; 2020211966@nwnu.edu.cn (L.S.); haozhj@nwnu.edu.cn (Z.H.); 2018211723@nwnu.edu.cn (X.S.)

[2] Gansu Province Internet of Things Engineering Research Center, Lanzhou 730070, China

\* Correspondence: dangxc@nwnu.edu.cn

**Abstract:** With the continuous development of intelligent transportation system technology, vehicle users have higher and higher requirements for low latency and high service quality of task computing. The computing offloading technology of mobile edge computing (MEC) has received extensive attention in the Internet of Vehicles (IoV) architecture. However, due to the limited resources of the MEC server, it cannot meet the task requests from multiple vehicle users simultaneously. For this reason, making correct and fast offloading decisions to provide users with a service with low latency, low energy consumption, and low cost is still a considerable challenge. Regarding the issue above, in the IoV environment where vehicle users race, this paper designs a three-layer system task offloading overhead model based on the Edge-Cloud collaboration of multiple vehicle users and multiple MEC servers. To solve the problem of minimizing the total cost of the system performing tasks, an Edge-Cloud collaborative, dynamic computation offloading method (ECDDPG) based on a deep deterministic policy gradient is designed. This method is deployed at the edge service layer to make fast offloading decisions for tasks generated by vehicle users. The simulation results show that the performance is better than the Deep Q-network (DQN) method and the Actor-Critic method regarding reward value and convergence. In the face of the change in wireless channel bandwidth and the number of vehicle users, compared with the basic method strategy, the proposed method has better performance in reducing the total computational cost, computing delay, and energy consumption. At the same time, the computational complexity of the system execution tasks is significantly reduced.

**Keywords:** internet of vehicles; mobile edge computing; edge-cloud collaboration; deep deterministic policy gradient networks; computational offloading

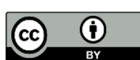

## 1. Introduction

In recent years, the rapid development of Internet of Things (IoT) technology, artificial intelligence, and big data has made people's requirements for travel services more complex and diverse [1,2]. According to statistics, the number of people using all types of vehicles is expected to reach 2 billion by 2035 [3]. In response to this environment, technologies such as intelligent mobility, autonomous driving, and the Internet of Vehicles (IoV) have emerged [4,5]. Autonomous vehicle technology allows multiple intelligent applications to be run, enabling vehicles to be driven more safely and efficiently [6]. Advanced communication and information technology are integrated into the IoV, which helps solve various traffic and driving problems, thus improving service quality needs such as passenger safety. Vehicular communication, in-vehicle communication, and vehicular mobile internet are the three main communication components of IoV [7]. Any latency in processing, analyzing, and collecting data in real-time is intolerable for IoV.

Cloud computing, fog computing, and Mobile Edge Computing (MEC) are a few intelligent computing platforms for big data analytics that can speed up data processing and effectively reduce latency [8-13].

Even some vehicles do not have the storage and computing capacity due to the limitations of the vehicle terminals [14]. In order to handle the huge amount of data collected through IoV, cloud computing is one of the ways to process this data efficiently [15]. The user in IoV is offloading the task requirements to the cloud computing center, which passes the processed results back to the user side. However, the distance between the cloud computing center and the users is long, and multiple users releasing service requests to the cloud computing center at the same time can cause a large delay, making it impossible for the users to receive the results of the cloud computing center promptly. Moreover, the vehicle is mostly on the move, which places higher demands on latency. Once the response latency exceeds the minimum requirement for a service request, it will likely cause safety problems and increase the risk of road accidents [16]. As a result, the bandwidth and latency of cloud computing are not suitable for processing real-time tasks in vehicles. Mobile edge computing is more promising than cloud computing as it solves the problem of insufficient computing power at vehicle terminals and allows for real-time data processing. Deploying MEC servers to a Road Side Unit (RSU), bringing computing resources closer to the side of the connected car user and sinking them to the MEC server, thus reducing latency to meet the changing needs of the vehicle user [17,18].

Recently, researchers have started using emerging techniques to solve computational offloading and resource optimization problems, such as artificial intelligence, machine learning, and reinforcement learning [19,20]. A great deal of research has shown that deep reinforcement learning-based approaches are widely used in many research areas and are more stable and efficient in solving dynamic computational offloading and resource optimization.

On the Internet of Vehicles mobile edge computing environment, the network environment, the needs of mobile vehicle users, and system computing resources are all changing. In a resource-constrained environment, using the distributed features of mobile edge computing can provide users with services with low latency and low energy consumption more stably and reduce the system's total cost. The current IoV mobile edge computing service offloads a significant challenge. Most of the existing research content is the research on computing task offloading based on the combination of edge and end. For this reason, this paper proposes a computing offloading method for the Internet of Vehicles based on Edge-Cloud collaboration. The contributions of this paper are as follows:

(1) In order to solve the problem of limited MEC resources, a task offloading overhead model based on Edge-Cloud collaboration based on multiple mobile vehicle users and MEC servers is designed. The model mainly has a three-layer architecture: the cloud service layer, the edge service layer, and the terminal layer. The method deployed on the MEC server is used to make real-time offloading decisions, and the service offloading method is partially optimized. The cloud center is responsible for coordinating the resources of the entire link.

(2) Within the system model, we formulate reducing the total cost of processing tasks by the system as an optimization problem, transforming the task offloading and resource optimization problem into an optimization problem based on deep reinforcement learning. Drawing on the idea of deep reinforcement learning, combined with the experience replay of DQN and the characteristics of the target network, an Edge-Cloud collaborative, dynamic computing unloading method (ECDDPG) based on a deep deterministic policy network is proposed.

(3) The simulation results show that the ECDDPG method can effectively reduce the computational complexity of system execution tasks. Even when the number of users reaches 35, compared to the baseline method, it can save 7.9–46.8% of the total cost.

The remaining chapters of this paper are organized as follows. The second section mainly summarizes the related work of this paper. The third section introduces the system model, communication model, related computing models, etc. The fourth section introduces the MEC offloading method on the Internet of Vehicles environment relatively thoroughly. Section 5 carries out the simulation experiment design and result from the analysis of the method in this paper. Section 6 concludes with a comprehensive summary of this paper.

## 2. Related Work

Nowadays, smart transportation is constantly evolving, and the scope of the Internet of Things is becoming more and more widespread, which continues to drive the growth of users' demand for computing [21]. With the advent of the MEC technique, the issue of efficiency and latency in data processing has again received increasing attention, with the problem of computational offloading of the MEC being widely followed by scholars.

In order to reduce the energy consumption of the MEC system to perform tasks, in [22], a heterogeneous two-layer computing offloading framework has been proposed, which can formulate joint offloading and multi-user association problems for multi-user MEC systems and effectively reduce the energy consumption of system processing tasks. Reference [23] proposes a new adaptive offloading algorithm, considering that a large number of resources of mobile devices are underutilized, as well as the spatiotemporal dynamics of devices, the uncertainty of service request volume, and the changes in the communication environment within the MEC system. Reference [24] builds a multi-user MEC system model under channel interference for continuous task execution and data partition-oriented applications. On this basis, the author proposes and solves the corresponding energy consumption minimization problem.

Recently, some scholars have studied the optimization of the MEC calculation offloading problem in the MEC system with the goal of reducing the system execution delay. The literature [25] considers the problems of offloading decisions, collaborative relay selection, and resource allocation among multiple users. It proposes a joint iterative algorithm based on Lagrangian Dual Analysis, monotonic optimization algorithms, to minimize the execution delay of all tasks [26]. In the device-to-device MEC scenario, the computing tasks generated by the device can be offloaded to the edge server for computing or offloaded to nearby devices. In order to effectively reduce the system delay, consider the energy consumption and partial offloading required to perform tasks, and resource allocation constraints, a new scheme based on joint partial offloading and resource allocation is proposed.

The above research work only considers the energy consumption or execution delay of the system executing tasks. Reference [27] studies the safe offloading framework of the mobile edge under Unmanned Aerial Vehicle (UAV) in a MEC network, describes it as a multi-objective optimization problem, and proposes a multi-objective optimization strategy based on the DQN algorithm, which can better reduce delay and reduce energy consumption. Reference [28] studies a distributed machine learning method based on a multi-user MEC network in a cognitive eavesdropping environment, proposes three optimization criteria, and uses a federated learning method to solve combinatorial optimization problems. Reference [29] proposes an adaptive task offloading and resource allocation method in the MEC environment, using deep reinforcement learning to select appropriate task computing nodes for mobile users, which can optimize the average response delay of the task and the total energy consumption of the system.

In the IoV environment, MEC can meet the low-latency and diverse requirements of mobile vehicle users. In recent years, some researchers have received extensive attention and research on applying MEC in IoV. Reference [30] designs a model framework combining MEC and IoV, where all vehicle users and MEC servers within the model can act as offloading nodes. A task offloading method with task classification and mobile offloading nodes is proposed. In the literature [31], in the IoV environment, a layered architecture

is constructed to minimize the system's total time delay, a hybrid nonlinear programming optimization problem is established, and an online multi-decision optimization is established algorithm based on Lyapunov is proposed [32] to solve the service request problem in in-vehicle networks by jointly optimizing task offloading decisions and resource optimization problems, proposing a cloud-edge coordinated computational offloading scheme, designing a distributed optimization method based on this scheme, and finally obtaining the optimal solution.

In addition, regarding the challenges of computing offloading and resource optimization of in-vehicle edge computing, some researchers use deep reinforcement learning to solve the MEC computational offloading problem. The literature [33] models task offloading and propose a mobility-oriented computing offloading retrieval protocol for in-vehicle edge multiple access to improve service quality. The literature [34] proposes a deep reinforcement learning-based offloading method in a multi-cell vehicle network scenario, which is used to optimize communication and computational resources and reduce the energy consumption and latency of the system in performing computational tasks. Reference [35] considers a system model with multiple mobile users and multiple MEC servers in vehicular networking, combines deep reinforcement learning methods and improves the traditional Q-learning approach, and demonstrates through simulation experiments that the approach can achieve reduced energy consumption at different wireless bandwidths.

The above research has done related work in reducing energy consumption and system execution delay in the MEC environment of IoV. The current research work considers a single problem. The problem of limited MEC resources will be exposed in the scenario of multiple vehicle users. It cannot meet the task requests from multiple vehicle users simultaneously. Therefore, providing a reliable service for multi-vehicle users still faces significant challenges in the IoV mobile edge computing environment.

Regarding the issue above, in the IoV environment where vehicle users race, this paper designs a three-layer system task offloading overhead model based on the Edge-Cloud collaboration of multiple vehicle users and multiple MEC servers. Using the mighty computing power of cloud services to coordinate the calculation of the full link can effectively alleviate the problem of limited computing resources of the MEC server. Based on the consideration of different devices, computing devices at different levels undertook task calculations with different computing power requirements. Combined with the idea of deep reinforcement learning, an Edge-Cloud collaborative, dynamic computing offloading method (ECDDPG) based on a deep deterministic policy network is proposed. The method can make quick offloading decisions for task requests and effectively meet the user's real-time requirements for task processing. The simulation results show that this method can achieve the expected goals well.

## 3. System Model

In the IoV environment, this section establishes a collaborative Edge-Cloud task offloading overhead model based on multiple mobile vehicle users and multiple edge servers, with a system model consisting of a cloud computing center and multiple edge server roadside units, and individual mobile vehicle users. On this basis, the communication model of the system is first established by mathematical modeling to ensure the reliability of data transmission within the system. Then, the computational offloading Model is introduced in terms of local computing, MEC server computing, and cloud server computing. The task offloading problem for vehicle user computing on Internet of Vehicles mobile edge computing is abstracted as a complex optimization problem. The task offloading overhead model for Edge-Cloud collaboration is shown in Figure 1.

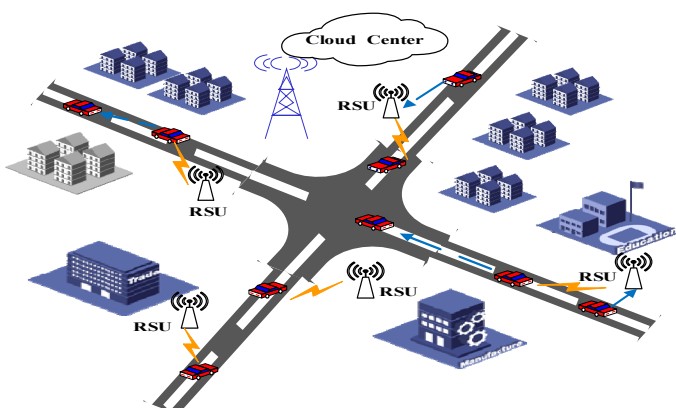

**Figure 1.** System Model.

### 3.1. Edge-Cloud Collaborative Mobile Edge Computing System Model

We consider a system model with multiple mobile users and MEC servers, where the locations of the cloud computing center and the MEC computing nodes are fixed, and each mobile vehicle user is arbitrarily mobile. At the same time, each mobile vehicle user has access to each MEC computing node, and each mobile vehicle user is relatively independent. The system model is mainly divided into three layers: the cloud service layer, edge service layer, and terminal layer. The cloud service layer makes use of the powerful computing and storage capabilities possessed by the cloud computing center, coordinates the computational power and intelligent resources on the entire link, gives full play to the advantages between different devices, and sends transmission instructions to the edge server when necessary, comprehensively providing services for end-vehicle users. The edge service layer consists mainly of MEC servers, which may consist of base stations, wireless access points or lightweight servers, and roadside units responsible for collecting information from vehicle users. There are multiple MEC nodes within the system MEC server, and the most appropriate node is selected to provide computing services to the vehicle users based on their request location. The terminal layer mainly consists of mobile vehicle users generating task requests or performing a limited number of task calculations. Considering the uncertainty of the wireless channel and the competition for server resources, minimizing the total system cost also requires solving the problem of system communication and the rational allocation of server computing resources.

There is one cloud center server in the system, $K$ MEC server can be represented as $\mathbb{k} = (1, 2, ...K)$, each MEC server has computing power, and mobile vehicle users are connected to the MEC server, $N$ mobile vehicle users can be represented as $\mathbb{N} = (1, 2, ..., N)$. The system time is defined as $t = (1, 2, ...T)$. Define the tasks generated by the mobile vehicle users $n$ as $D_i = (c_i, d_i, T_{max})$, where $c_i$ indicates the size of the period required to complete the calculation task, $d_i$ indicates the amount of data generated by the vehicle user. $T_{max}$ indicates the maximum time delay allowed for system processing tasks.

### 3.2. Communication Model

In the IoV, each vehicle user generates multiple independent tasks, which are divided into multiple sub-tasks with data dependencies. Based on the actual needs of the users, the tasks can either be computed locally, offloaded to the MEC server at the edge service layer for computation, or offloaded to a cloud server to be processed using a cloud center. Therefore, the costs under these three different computation methods are to be considered.

In this paper, the communication between the mobile vehicle user $n$ and the MEC computing node $k$ adapts to the wireless network. In order to better realize bidirectional

data transmission between users and edge nodes, users and MEC computing nodes adopt the orthogonal multiple access technology of frequency division multiplexing, and the bandwidth of each sub-channel is $W$. In a certain dynamic time-varying ideal channel state, the communication between each user and the MEC node will not be disturbed. Then, the maximum data transmission rate between the mobile vehicle user and the corresponding MEC node can be expressed as:

$$v_i(n,k) = m_i^c W \log_2\left(1 + P_i g_i / \sigma^2\right) \tag{1}$$

where $m_i^c$ indicates the number of wireless sub-channels allocated by the system to the vehicle user $n$, $\sigma^2$ represents the background noise power of the Internet of Vehicles environment, $P_i$ represents the vehicle user $n$ transmit signal power. $g_i$ represents the channel gain between the vehicle user $n$ and the MEC node $k$.

### 3.3. Computational Offloading Model

For the computing-intensive task request of each vehicle used on the Internet of Vehicles, the task can be selected for local computing, offloaded to MEC server computing, or offloaded to the cloud server computing center. Therefore, the vehicle user's choice of the appropriate offloading strategy plays a crucial role in the system function. The purpose of the offloading strategy is to face the dynamic offloaded of multiple mobile vehicle users, reduce the delay and energy consumption of computing tasks, improve the computing speed, use resources reasonably, and better meet the needs of users. For the offloading decision parameter, we use a 0–1 variable, indicating as $\alpha$, $\beta$, when $\alpha_i = 0$. This means that the mobile vehicle user selects their CPU to process the generated data. When $\alpha_i = 1$, the tasks generated by the mobile vehicle user need to send an offloading request to the edge server, making the appropriate decision for the offloading. If $\beta_i = 0$, then the task data is offloaded to the MEC server for calculation; If $\beta_i = 1$, then the task data will be offloaded to the cloud server for computational processing. This is represented as follows:

$$\alpha_i = \begin{pmatrix} 0, & \text{Local Computing} \\ 1, & \text{Others} \end{pmatrix}, \quad \beta_i = \begin{pmatrix} 0, & \text{MEC Server Computing} \\ 1, & \text{Cloud Server Computing} \end{pmatrix} \tag{2}$$

The tasks generated by mobile vehicle users can be processed locally and offloaded to edge servers or cloud servers for processing. Therefore, we discuss the latency, energy consumption, and total consumption cost arising from local computing, edge server computing, and cloud server computing.

### 3.3.1. Local Computing

The task data generated by the mobile vehicle user is calculated locally, and the calculation process is only related to the CPU computing capability of the vehicle user. In our scenario, for tasks performed by local computation, let $f_i^{loc}$ denote the computing power of the *i*-th end vehicle user.

In heterogeneous IoV, there is no transmission delay for local computation. Therefore, the delay caused by the user of the mobile vehicle performing the task locally can be expressed as:

$$T_i^{loc} = c_i / f_i^{loc} \tag{3}$$

According to the literature [22], let the energy consumption model be as $\kappa_i \left(f_i^{loc}\right)^3$, then the energy consumption generated by the mobile vehicle user processing computing tasks at the terminal layer is expressed as follows:

$$E_i^{loc} = \kappa_i \left(f_i^{loc}\right)^3 T_i^{loc} \tag{4}$$

where $\kappa$ depends on the chip structure of the terminal device in the system, let $\kappa = 10^{-26}$. According to Formulas (3) and (4), we can obtain the total cost required to execute the task data locally:

$$\mathbb{Q}_i^{loc} = \varepsilon^{loc} T_i^{loc} + \varphi^{loc} E_i^{loc} \tag{5}$$

The sum of coefficients $\varepsilon^{loc}$ and $\varphi^{loc}$ in Formula (5) represents the time weight and energy consumption weight generated by the user performing the calculation task locally.

### 3.3.2. MEC Server Computing

For tasks performed by the edge server, the execution of the task generates additional latency and energy overhead so that vehicle users can wirelessly transmit data to the edge. The total delay generated by mobile vehicle users performing computing tasks on the edge server comprises four parts: transmission delay, calculation delay, waiting for the delay, and result return delay. According to the relevant literature and data results, the delay caused by the result back to the terminal layer is slight, which is ignored in this paper. Let $f_i^{MEC}$ denote the computing power of the $i$-th edge server, according to the Formula (1) of the communication model. Therefore, in the heterogeneous IoV, the delay generated by the sub-tasks generated by different vehicle users in the execution of tasks on the edge server can be expressed as:

$$T_e^{com} = \frac{c_i}{f_i^{MEC}} \tag{6}$$

According to Formula (1), the time delay consumed by the task data generated by the mobile vehicle user transmitted from the terminal layer to the edge server through the wireless channel is expressed as:

$$T_e^{tran} = \frac{d_i}{v_i(n,k)} \tag{7}$$

Therefore, if the data task request of a mobile vehicle user is uninstalled to the edge server for computation, the resulting total delay can be expressed as:

$$T_e^{edge} = T_e^{com} + T_e^{tran} + T_e^{wait} \tag{8}$$

Among them, $T_e^{wait}$ is the queuing delay for the user to make a task request. The queuing delay for any user is the time interval from when the user sends a request to the system to when the task is executed. $T_e^{sta}$ denotes the start time of the task request, $T_e^{end}$ denotes the time when the task request starts to be executed.

$$T_e^{wait} = T_e^{end} - T_e^{sta} \tag{9}$$

When the data task request of the mobile vehicle user is offloaded to the edge server to perform the calculation, the energy consumption is determined by the data transmission in the channel, the time the system waits, and the energy generated by the calculation at the edge server. The total energy consumption is expressed as:

$$E_e^{edge} = P_i^N (T_e^{tran} + T_e^{wait}) + \kappa_i \left( f_i^{MEC} \right)^3 T_e^{com} \tag{10}$$

According to the above formula, the total cost of vehicle users performing computing tasks on the edge server is:

$$\mathbb{Q}_i^{edge} = \varepsilon^{edge} T_e^{edge} + \varphi^{edge} E_e^{edge} \tag{11}$$

The sum of coefficients $\varepsilon^{edge}$ and $\varphi^{edge}$ in Formula (11) represents the time weight and energy consumption weight generated by performing the calculation task on the MEC server.

### 3.3.3. Cloud Server Computing

The MEC server cannot process the data tasks generated by the mobile vehicle users faster and is offloaded to the cloud server for processing through the wireless network in the channel. At this time, the total delay generated by the cloud server processing computing task is mainly composed of the transmission delay generated by the user passing the task to the cloud server layer, the delay generated by the cloud server processing computing task, the waiting delay of the mobile vehicle user and the delay generated by the task result back to the mobile vehicle user. The delay caused by the return of the cloud processing task results to the terminal layer is small and can be ignored. The overall latency generated by cloud server computing can be expressed as:

$$T_c^{cloud} = T_c^{tran} + T_c^{com} + T_c^{wait} \tag{12}$$

The transmission delay is expressed as:

$$T_c^{tran} = \frac{d_i}{v_i(n,k)} \tag{13}$$

The delay of data tasks generated by mobile vehicle users in cloud server computing is expressed as:

$$T_c^{com} = \frac{c_i}{f_i^{cloud}} \tag{14}$$

When the data task request of mobile vehicle users is offloaded to the cloud server for computing, the energy consumption is generated by data transmission in the channel, system waiting time and computing in the cloud server. Thus, the total energy consumption incurred in processing task data at the cloud server is expressed as:

$$E_c^{cloud} = P_i^N (T_c^{tran} + T_c^{wait}) + \kappa_i \left( f_i^{cloud} \right)^3 T_c^{com} \tag{15}$$

From the above formula, the overall cost of vehicle users performing computing tasks on the cloud server can be expressed as:

$$\mathbb{Q}_i^{cloud} = \varepsilon^{cloud} T_c^{cloud} + \varphi^{cloud} E_c^{cloud} \tag{16}$$

The sum of coefficients $\varepsilon^{cloud}$ and $\varphi^{cloud}$ in Formula (16) represents the time weight and energy consumption weight generated by performing the calculation task on the cloud server.

### 3.4. Problem Formulation

The purpose of the offloading decision is to reduce the system's total cost to complete the mobile vehicle user task request as much as possible. If the task requests of all mobile vehicle users are executed locally, the MEC server and the cloud server are idle; if the task requests are offloaded to the MEC server, the computing resources of the vehicle users and the cloud server are in an idle state.

According to the above analysis, the minimum cost problem of the system can be expressed as:

$$\text{Min:} \mathbb{Q}_i^{system} = \alpha_i \mathbb{Q}_i^{loc} + (1 - \alpha_i)(\beta_i \mathbb{Q}_i^{edge} + (1 - \beta_i)\mathbb{Q}_i^{cloud})$$

$$\text{S.t.:}$$

$$3a: T_c \leq T_{max}$$

$$3b: T_e^{edge} \leq T_{max} \tag{17}$$

$$3c: \alpha + \beta = 1, \alpha_i \in [0,1], \beta_i \in [0,1]$$

$$3d: \varepsilon + \varphi = 1$$

$$3e: \text{Formula (5) (11) (16)}$$

Among them, the constraints 3a and 3b are that the delay generated by the local task execution and the MEC server execution task cannot be higher than the maximum delay of the system. Therefore, 3c indicates the relationship between the decision parameters of the system offloading, 3d indicates the relationship between the two weight coefficients, and 3e is determined according to the calculation model of the system. The total cost is incurred by the system to perform a computing task.

Since wireless channels and vehicle users are dynamically changing and moving, traditional optimization problems cannot solve the resource optimization problem better. Resource allocation decisions and unloading decisions are related to the current system utility and affect the state of subsequent system processing tasks. To this end, we combine the methods of deep reinforcement learning to solve the above problems. See the next chapter for details.

## 4. Dynamic Computing Offloading Method Based on Edge-Cloud Collaboration

Section 3 derives the total cost of executing a system's task requests under different computational offloading models, and this section considers a collaborative Edge-Cloud-based offloading approach. For the offloading problem of multiple edge servers and multiple mobile vehicle users in IoV, the information obtained by the RSU, such as the size of the service requested by the vehicle user, the transmit power of the user, the channel gain, etc., in order to minimize the total cost of the system to handle the task. Combined with a deep reinforcement learning method for offloading decisions, an Edge-Cloud collaborative, dynamic computing offloading method (ECDDPG) based on Deep Deterministic Policy (DDPG) is proposed. The method can perform independent learning at each MEC server and select an optimal action for each vehicle user according to the interaction between the agent and the environment, i.e., the best choice for the system's task execution.

### 4.1. Introduction to the DRL

Some definition standards of DRL mainly refer to traditional RL methods, including an agent, a set of environmental state spaces $S$, and a set of action spaces $A$. The agent makes corresponding action decisions through continuous interaction with the environment in discrete time $t$. At each time $t$, the agent observes the current state $S_t$ of the system from the state space $S$, and chooses an appropriate action from the action space $A$ according to a random policy $\pi$, i.e., $\pi : S \rightarrow P(A)$, Then, the agent gets an instant reward $r_t = (s_t, a_t)$, transition to the next state $S_{t+1}$ according to the transition probability $P(s_{t+1} \mid s_t, a_t)$ of the environment, each action performed will receive a reward from the environment. In order to find the optimal strategy, when the long-term cumulative return benefit reaches the maximum, it is the optimal strategy in the current state. The environment in which the system is located is defined according to the sum of discount rewards in a particular state is:

$$R_t = \sum_{i=t}^{T} \gamma^{(i-t)} r(s_i, a_i) \tag{18}$$

where $T$ is the total time step of the system, $\gamma \in (0,1)$ represents the discount factor. The main goal of RL is to find a strategy in the initial distribution state to maximize the reward $R$ with discount accumulation, the formula is expressed as:

$$J = \mathbb{E}_{s_i \sim E, a_i \sim \pi}[R_t] \tag{19}$$

---

**Algorithm 1** ECDDPG-based Dynamic computational offloading algorithm

---

Input: Sets of vehicular users, MEC services $K$, task $D_i$,

Output: Maximum Reward $R$ and Action $A$;

    1: Initialize weights of actor and critic online networks, $\theta^\mu$ and $\theta^Q$;

    2: Initialize weights of actor and critic target networks, $\theta^{\mu'}$ and $\theta^{Q'}$;

    3: Initialize experience replay buffer $\mathcal{B}$ to be empty;

4: **For** each episode = $1, 2, \cdots \mathcal{E}$ **do**

5:    Reset simulation parameters for the Internet of Vehicles environment;

6:    Randomly generate an initial state $s_t \in S$;

7:    **For** each time slot = $t = 1, 2, \cdots T_{\max}$ **do**

8:    Select an action $a_t = \mu(s_t \mid \theta^\mu) + N_t$, Decide whether vehicle user tasks are to be performed locally or computation offloading offload according to the current policy and noise;

9:    Execute the action $a_t$, receive an award $r_t \in R$, transition to the next state $s_{t+1}$;

10:    Store transition $(s_t, a_t, r_t, s_{t+1})$ into experience replay buffer $\mathcal{B}$;

    11:    Sample a random mini-batch of $\chi$ transitions $(s_i, a_i, r_i, s_{i+1})$ from the buffer $\mathcal{B}$;

12:    Update the parameters of critic network by minimizing the loss function based on equation (24);

13:    Update the parameters of actor network by using the policy gradient based on equation (26);

14:    Update the target network: $\theta^\mu, \theta^Q$ based on equation (27) and (28);

15:    end for

16: end for

---

### 4.2. Dynamic Offloading Decision Method Based on DDPG

#### 4.2.1. Problem Transformation

In the IoV environment, the vehicle user is in a state of high-speed movement; this will make the channel gain, wireless channel state, vehicle user MEC server, and cloud server state in this system model dynamic and variable. The system needs to make corresponding unloading decisions at different time nodes, allocate system resources more reasonably, and achieve the goal of reducing the total cost. We know that traditional dynamic programming algorithms are effective in solving such problems. However, we cannot ignore that dynamic programming algorithms require a significant computational cost to perform the task, which requires a list of all possible resource allocation scenarios and unloading strategies before choosing an optimal offloading strategy. Therefore, it is not easy to make real-time decisions for vehicle users using a dynamic programming algorithm, and the reinforcement learning method can make real-time decisions according to different states of users.

The agent in the RL method continuously interacts with the Internet of Vehicles environment to obtain corresponding rewards, optimizes the strategy according to the reward value, and finally finds an optimal resource allocation strategy. Therefore, we transform the problem in Equation (17) into an optimization problem for DRL. The traditional DQN method mainly deals with discrete, discontinuous actions, which is unsuitable for this paper's scene. Compared with the DQN method, DDPG is more suitable for solving continuous action problems. For computational offloading and resource optimization problems, the agent, state space $S$, action space $A$, and reward function $R$ are respectively defined as follows:

Agent: In our method, the agent chooses an action by interacting with the environment and receives a corresponding reward, which continuously transitions to the next

state over time until the end of training. If the agent can satisfy the constraints (3a) ~ (3e), the system model will give a positive reward; otherwise, it will be punished accordingly and receive a negative reward.

State space: According to the objective of minimizing the total cost of the system execution task, the total cost of the delay and energy consumption generated by the execution task data is regarded as a state, which represents the system state space set. The state indicates the total cost of the task in local computing, MEC server computing, and cloud server computing. The state-space can be defined as:

$$s_t = \left[ \mathbb{Q}_i^{loc}, \mathbb{Q}_i^{edge}, \mathbb{Q}_i^{cloud} \right] \in S \tag{20}$$

Action Space: This paper considers the problem of task offloading and resource optimization in the MEC environment of the Internet of Vehicles. The offloading decision will judge whether the task requested by the mobile vehicle user is to be offloaded, and the offloading decision is regarded as an action selection. In this system model, the mobile vehicle user, the MEC server, and the cloud center have computing resources to perform calculations, which can be used as the object of action selection. Therefore, the action space can be expressed as follows:

$$a_t = \left[ a_{n,1}(t), ..., a_{n,i}(t), a_{k,1}(t), ...a_{k,i}(t), cloud \right] \in A \tag{21}$$

Obviously, the action set $a_t$ is an explicit policy related to the vehicle user $n$, which is all possible actions that the vehicle user can select in time, and it is a continuous action problem. When the action space chooses to assign 1 to the action value at the MEC server, the rest of the computing nodes are all 0. Assuming that the task is computed on the MEC server, the action can be expressed as follows:

$$a_t = \left[ 0, ..., 0, 1, ...1, 0 \right] \tag{22}$$

Reward function: In the current state, an action is selected as the best offloading decision action. If the task request needs to be offloaded to the MEC server or cloud center for computing; otherwise, the task is executed on the mobile vehicle. In order to better improve the performance of the network, the agent interacts with the environment to obtain corresponding rewards. Our goal in this paper is to minimize the total cost of the system, where $\ell$ is the coefficient factor, and the reward function is expressed as follows:

$$r_t(s_t, a_t) = -\ell \mathbb{Q}_i^{system} \tag{23}$$

### 4.2.2. Dynamic Offloading Decision Method

The Deep Deterministic Policy Gradient Networks (DDPG) [36] method is different from the traditional DQN method [37], which can only deal with low-dimensional and relatively discrete action spaces. For some action sets, it may be a continuous action value or a very high-dimensional discrete value. The DQN network relies on finding the optimal solution of the action-value function in each iteration. For continuous sets of action spaces, the DQN networks lack processing power, and their trained models cannot cope with random strategies. Policy-based RL methods can better meet this challenge, but these methods can deal with continuous action space aspects based on learning deterministic or stochastic policies. However, these methods still show the problem of too slow convergence speed, based on the deterministic policy gradient (DPG) method [38] to approximate the action value gradient with the action value (Q) function. DDPG is relatively simple in that it combines the features of policy-based and value-based methods to deal with the continuous action space without outputting the probabilities of the actions and directly outputting the magnitude of the values of each dimension corresponding to the actions. To this end, we propose an Edge-Cloud collaborative dynamic offloading method (ECDDPG) based on DDPG. Figure 2 shows the ECDDPG method architecture. ECDDPG inherits the advantages of traditional DQN methods, such as experience replay and the

target network. The structure of the Actor and Critic network uses DNN, which contains two networks, namely Actor online network, Actor target network, Critic online network, and Critic target network.

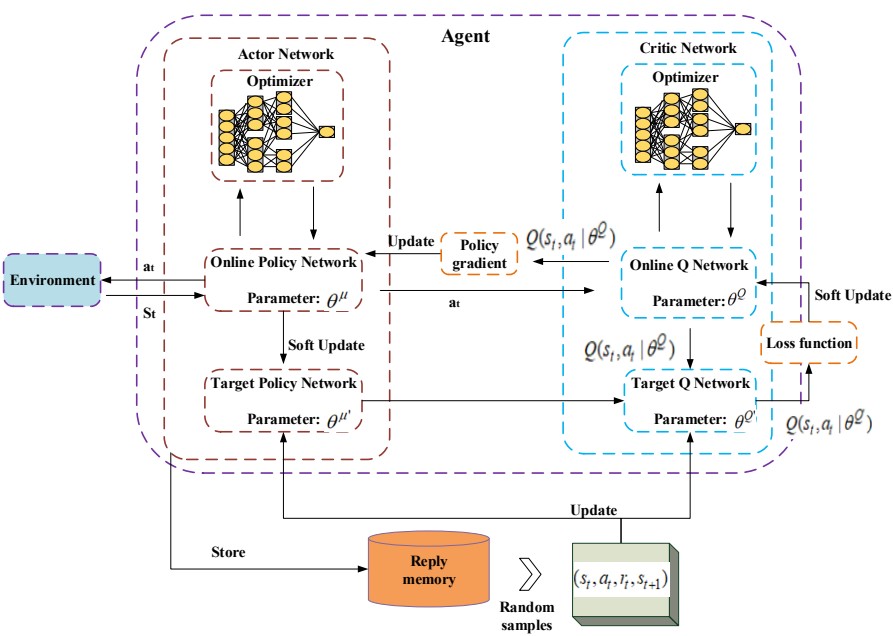

**Figure 2.** ECDDPG Method Structure Diagram**.**

The specific steps are shown in Algorithm 1. We take the number of vehicle users, the number of MEC servers, and the tasks requested by the users as input, and take the maximum reward and optimal offloading policy as the output. First, steps 1–3 initialize the various parameters $\theta^\mu, \theta^{\mu'}, \theta^Q, \theta^{Q'}$ of Actor and Critic network, as well as the replay buffer $\mathcal{B}$. Step 4 starts training the intelligence in the environment we have designed for $\mathcal{E}$ iteration, and the MEC server can learn the determined service policy independently after $\mathcal{E}$ iteration. Second, we perform parameter simulation for the Internet of Vehicles environment in steps 5 and 6, and randomly generate a state $s_t \in S$, which is set as the initial state. Steps 7~11 start training within a time period. For the training period, the current strategy network $\theta^\mu$ and noise $N_t$ are executed. Select an action $a_t = \mu(s_t \mid \theta^\mu) + N_t$ in step 8 to decide where to offload. As the number and location of users change, to reduce the cost of method retraining and the computational complexity, adopt an order-preserving quantification approach within the ECDDPG method. Step 10 stores the transformation tuple $(s_t, a_t, r_t, s_{t+1})$ in the buffer. When the buffer memory is full, the Critic network uses the experience replay technology. In the time slot, replaying the records $(s_t, a_t, r_t, s_{t+1}) \sim \mathcal{B}$ stored in the tuple requires random sampling of a small batch of data in the tuple for training and updating. Step 12 updates the parameters of the Critic network by minimizing the loss function. The minimum loss function in the judging process can be expressed as:

$$L(\theta^Q) = \frac{1}{\chi} \sum_{t=1}^{\chi} \left[ (Q(s_t, a_t \mid \theta^Q) - y_t)^2 \right] \tag{24}$$

Among them, the Q value of the target network is calculated by the Critic network; that is, it can be expressed as:

$$y_t = r_t + \gamma Q'(s_{t+1}, \mu'(s_{t+1} \mid \theta^{\mu'}) \mid \theta^{Q'}) \tag{25}$$

In the DPG approach, parameterize $\mu^{'}(s_{t+1} \mid \theta^{\mu^{'}})$ is used to formulate the current strategy. Step 13 updates the parameters of the Actor network by using the policy gradient, which is expressed as:

$$\nabla_{\theta^{\mu}} J \approx \frac{1}{\chi} \sum_{t=1}^{\chi} \nabla_a Q^{'}(s_t, a \mid \theta^Q)|_{a=a_t} \, \nabla_{\theta^{\mu}} \mu(s_t \mid \theta^{\mu}) \tag{26}$$

Using soft update method to update the target network parameters:

$$\theta^{\mu'} \leftarrow \tau\theta^{\mu} + (1-\tau)\theta^{\mu'} \tag{27}$$

$$\theta^{Q'} \leftarrow \tau\theta^{Q} + (1-\tau)\theta^{Q'} \tag{28}$$

## 5. Simulation Design and Result Analysis

This chapter first introduces the simulation experiment environment, then briefly introduces the relevant parameter settings of the experiment, and finally proves the method's feasibility in this paper by designing experiments.

### 5.1. Simulation Environment and Parameter Settings

The ECDDPG method proposed in this paper is simulated and verified using Python3.7 and TensorFlow1.14 on a PC with Intel(R) Core (TM) i5-7400 3.0GHZ CPU processor and 8GB memory. The relevant vehicle network model parameters are set according to the IEEE 802.11p standard and 3GPP TR36.885. Five MEC servers and one cloud center server are selected, and an RSU with a MEC server is placed at regular intervals, each RSU covers 100 × 200 m2, and the cloud center covers the entire road.

As in reference [31], vehicle user transmit signal power is $P_i = 100mW$. We set the task data size and request content size to [0.15–0.8] GB. The channel gain between vehicle users and MEC nodes is $L[dB] = 128.1 + 37.5 \log 10(d[m])$. The soft update parameter is 0.01. The complete system simulation parameters are shown in Table 1.

**Table 1.** Simulation parameters.

| Parameter | Value | Parameter | Value |
|---|---|---|---|
| channel bandwidth | [5–30] | Vehicle computing power | $10^6$ |
| Number of vehicle users | [5–35] | Number of RSUs | 5 |
| MEC server computing power | $3 \times 10^8$ | Number of MEC servers | 5 |
| vehicle speed | 50, 60 km/h | Number of main lanes | 5 |
| Actor network learning rate | 0.005 | Critic network learning rate | 0.005 |
| Channel noise power | $10^{-10}$ | Number of iterations | 1000 |

### 5.2. Analysis of Simulation Results

In this section, the method in this paper is mainly compared with the following benchmark schemes: (1) ALL-Loc strategy, which calculates all tasks locally; (2) ALL-MEC policy, where all tasks are offloaded to the MEC server for execution; (3) Random offloading strategy, the task data generated by the moving vehicle is randomly offloaded to the MEC server or cloud server for calculation. The simulation experiment design mainly considers the influence of different methods of reward size, wireless channel bandwidth, and the number of vehicle users on the delay, energy consumption, and total cost of executing tasks.

#### 5.2.1. Performance Evaluation of Different methods

In order to verify the superiority of the performance of our proposed method, in the same IoV mobile edge computing environment, the maximum number of iterations is 1000. The size and convergence of the reward values of the proposed, DQN, and Actor-Critic methods are compared. As shown in Figure 3, the proposed method ECDDPG keeps the maximum reward

value as the number of iterations increases. When the number of iterations is less than 300, the reward value of the method proposed in this paper is slightly higher than that of the DQN method. However, the convergence of this paper is better, while the convergence of the Actor-Critic method is always unstable. When the number of iterations is greater than 300, the reward value of the method proposed in this paper is much higher than the other two methods, gradually converging to the optimum. In the face of continuous action space sets, the shortcomings of the DQN method are magnified. As the number of iterations increases, the reward value gradually becomes smaller and smaller. Therefore, it can be reflected that the ECDDPG method in this paper is superior to the DQN method and the Actor-Critic method in terms of convergence and reward value.

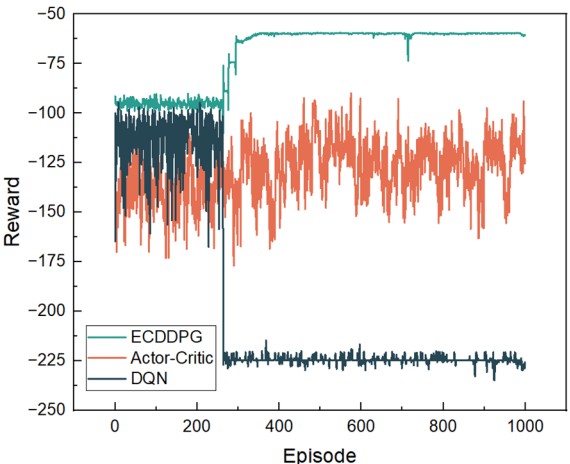

**Figure 3.** Reward and Convergence of Different Methods.

5.2.2. Influence of Wireless Bandwidth Change on System Performance

In Figure 4, when the number of users is 20, we consider the influence of the wireless channel bandwidth in the system on the time delay of each method in the execution of computing tasks. We know that when the task is decided to be executed locally, there is no need to transfer any task to the MEC server or cloud service to perform the calculation, and there is no transfer delay. Therefore, the delay caused by the ALL-Loc policy execution task is not affected by bandwidth changes. Since the task data generated by the vehicle user needs to be transmitted to each RSU through the wireless channel and then calculated by the MEC server, the delay generated by the ALL-MEC strategy is greatly affected by the bandwidth. The effect of the random offloading strategy is between our method and the ALL-MEC strategy. In general, no matter how the bandwidth changes, the method proposed in this paper can adapt well, and the delay generated by the task execution is lower than other methods.



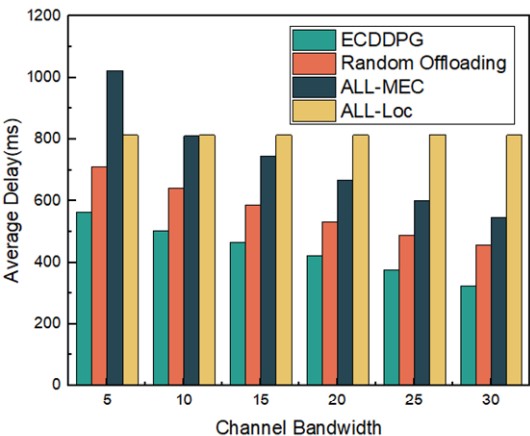

**Figure 4.** Influence of the Bandwidth Change on Time Delay.

In Figure 5, we verify what kind of change law the change of wireless bandwidth will bring to the system's energy consumption to perform tasks. According to the formula, we can see that the energy consumption generated by the system is proportional to the time. The energy consumption generated by the ALL-Loc strategy is unaffected by changes in wireless bandwidth, as is the time delay, and remains constant over a defined period. The ALL-MEC strategy is still the most affected by changes in wireless bandwidth. The random offloading strategy randomly offloads tasks to the MEC server or cloud server to perform computation. Therefore, the energy consumption change generated by it is less affected by the system bandwidth than the ALL-MEC and ALL-Loc strategies. However, the method in this paper is minimally affected by the change in wireless bandwidth, which further highlights the method's superiority in reducing energy consumption.

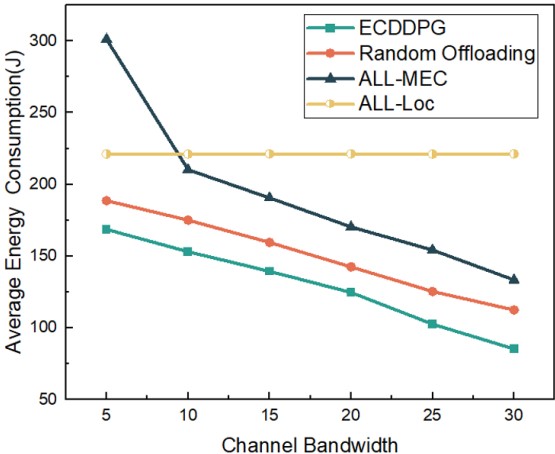

**Figure 5.** Influence of Bandwidth Change on Energy Consumption.

In Figure 6, we analyze the impact of changes in wireless bandwidth on the total system cost. The total cost in this paper is the weighted sum of energy consumption and delay. The total cost of ALL-Loc strategy execution is still not affected by the change of wireless bandwidth, while the total cost generated by the ALL-MEC strategy is still the largest. The variation of the machine offloading strategy lies between our method and the ALL-MEC strategy. Through continuous training and optimization, the method in this paper can better adapt to changes in bandwidth and output the optimal offloading strategy and the maximum reward value. Therefore, the method in this paper still shows great advantages in reducing the system's total cost.

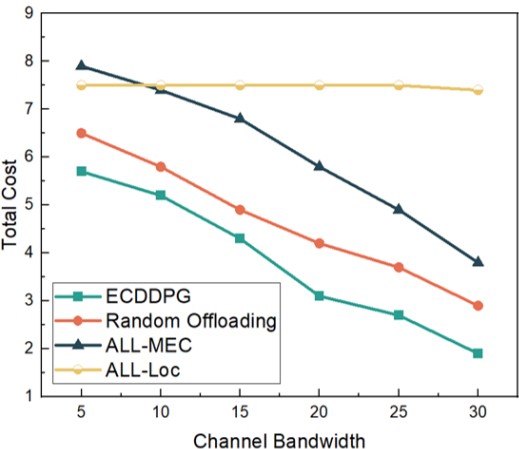

**Figure 6.** Influence of Bandwidth Change on Total Cost.

### 5.2.3. The Effect of the Number of Vehicle Users on Method Performance

In order to further verify that the algorithm in this paper can effectively reduce the delay caused by executing tasks, the channel bandwidth is set to 15MHz, and we consider the impact of changes in the number of vehicle users on the system processing task delay, energy consumption, and total system cost. Set the number of vehicle users to 10, 15, 20, 25, 30, and 35.

From Figure 7, we can see that with the increasing number of vehicle users, the delay in executing tasks will also increase. However, it is not difficult to see that the method in this paper is always optimal. Specifically, due to the limitation of MEC server resources, when the number of vehicle users reaches more than 20, the ALL-MEC strategy cannot process task data instantly, and the queuing time of task execution will increase sharply. Therefore, this strategy's total delay growth rate becomes faster and faster. The computing power of the local vehicle is much smaller than that of the MEC server, but the execution of the ALL-Loc strategy does not generate latency. Therefore, when the number of vehicle users is small, the delay caused by the ALL-MEC strategy is not apparent. The random unloading strategy can fully use cloud center resources, MEC servers, and local computing resources, and its delay is less than the above two methods. The increase in the number of vehicle users will not affect it much, but the delay is still higher than the method proposed in this paper. According to the change in the number of users and the number of tasks, the method in this paper can make the optimal unloading decision more quickly by interacting with the environment and constantly adapting to the external environment. Therefore, the superiority of the method proposed in this paper in reducing the delay can be effectively verified.

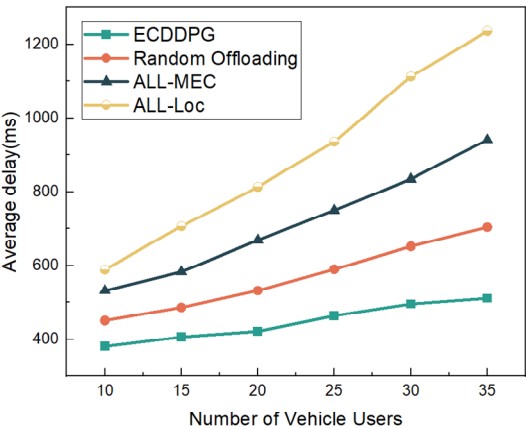

**Figure 7.** Influence of the Vehicle User Number on Time Delay.

Figure 8 shows the effect of changes in the number of vehicle users on energy consumption. As energy consumption and latency are proportional, changes in the number of vehicle users at a given time have the same effect on energy consumption as latency, both becoming more frequent as the number of vehicle users increases. In the case of our proposed method, the generated energy consumption is always the lowest, the method in this paper mainly considers the optimal offloading strategy, and one of the goals is to minimize the energy consumption. The energy consumption generated under the ALL-MEC strategy is relatively large. The more vehicle users, the more tasks are generated. The resources of the MEC server are limited, and all offloading to the MEC server will generate more energy consumption.

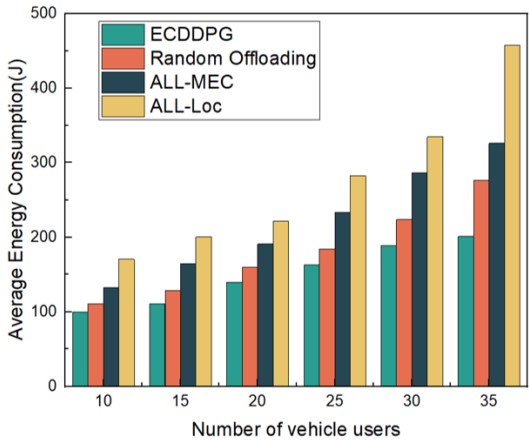

**Figure 8.** Influence of the Number of Vehicle Users on Computing Energy Consumption.

The effect of the change in the number of vehicle users on the system's total cost is illustrated in Figure 9. Increasing the number of users inevitably requires more time and energy consumption, increasing the corresponding total system cost. The other three methods vary more significantly than the method in this paper. The method in this paper mainly considers the optimal unloading action in a particular state, after optimal training, to output a more favorable reward, and through the accumulation of time, eventually output an optimal reward, and thus can minimize the total cost of completing the task. When the number of users is 35, the total cost of the ECDDPG method is reduced by 7.9%~46.8% compared with the other three methods. The computational complexity is effectively reduced, further verifying the method's superiority in this paper.

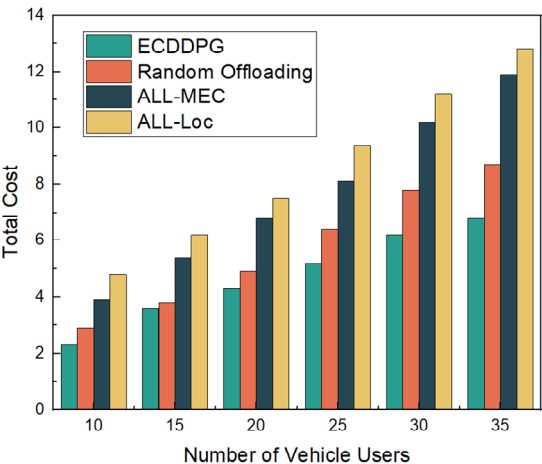

**Figure 9.** Influence of the Number of Vehicle Users on the Total Cost.

### 6. Conclusions

In the environment of mobile edge computing of the Internet of Vehicles, in order to solve the problem of minimizing the cost of system execution tasks, we designed a task offloading overhead model for the "Edge-Cloud" collaborative MEC system based on multiple mobile vehicle users and multiple edge servers, and introduced the communication model, computational offloading model, and problem modeling. We formulated reducing the total cost of system processing tasks as an optimization problem to minimize the total cost of vehicle users, MEC servers, cloud center, and the task offloading. The resource optimization problem is transformed into a combinatorial optimization problem based on deep reinforcement learning; combined with the idea of deep reinforcement learning, a dynamic computing offloading method based on Edge-Cloud collaboration based on a deep deterministic policy network is proposed (ECDDPG). The simulation results show that this paper's method effectively reduces the system execution cost.

There are still some shortcomings in the research of this paper, such as the vehicle resources parked on the roadside not being fully utilized, the task data is not divided into the result part, and the problem of complete unloading is considered. In the future, we will study the task offloading problem of the in-vehicle network, divide the task into multiple parts, and make full and reasonable use of the vehicle resources parked on the roadside.

**Author Contributions:** Conceptualization, X.D., L.S. and Z.H.; methodology, X.D and L.S.; software, L.S. and X.S.; validation, X.D., L.S. and Z.H.; formal analysis, X.D., L.S. and X.S.; investigation, Z.H.; resources, X.D. and Z.H.; data curation, X.D., L.S. and X.S.; writing—original draft preparation, L.S. and X.S.; writing—review and editing, X.D., Z.H. and X.S.; visualization, X.D. and Z.H.; supervision, X.D.; project administration, X.D. and Z.H.; funding acquisition, X.D. and Z.H. All authors have read and agreed to the published version of the manuscript.

**Funding:** This research was funded by National Natural Science Foundation of China (Grant 62162056), Industrial Support Foundations of Gansu (Grant No.2021CYZC-06) by X. D. and Z.H.

**Conflicts of Interest:** The authors declare no conflict of interest.

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
