# Peer review of "Dynamic Offloading Method for Mobile Edge Computing of Internet of Vehicles Based on Multi-Vehicle Users and Multi-MEC Servers"

_electronics, doi:10.3390/electronics11152326_

Round 1
Reviewer 1 Report
This paper presents a Dynamic offloading method for mobile edge computing of the Internet of Vehicles based on DDPG, which is based on all areas of IoT. Thus, this paper is directly related to the theme of this journal.
Overall, the paper is organized properly; So, the paper is accepted after the following major changes:
1. In the Introduction problem statement is not clear, why the researcher chooses this area for work, and what is the motivation behind solving the problem (which is not clearly defined in the introduction). Highlight this in a revised version
2. Figure 1 system model is a very simple picture of the vehicle cloud if protocols, communication details, and other components on each layer will be added then the figure will have more worth in sense of knowledge
3. Too long paragraphs in Related work which made it difficult to readability so divide into short paragraphs.
4. Figures 2 and 4 are not clear, redraw figures and provide clear images
5. add a few references related to IoT, which are mentioned below
Kaishan Wu, Rashid Ali Laghari, Mureed Ali, and Abdullah Ayub Khan. "A Review and State of Art of Internet of Things (IoT)." Archives of Computational Methods in Engineering (2021): 1-19.
Waqas, Muhammad, Kamlesh Kumar, Umair Saeed, Muhammad Malook Rind, Aftab Ahmed Shaikh, Fahad Hussain, Athaul Rai, and Abdul Qayoom Qazi. "Botnet attack detection in Internet of Things devices over cloud environment via machine learning." Concurrency and Computation: Practice and Experience 34, no. 4 (2022): e6662.
Huang, I., Yu-Hsuan Lu, Muhammad Shafiq, Asif Ali Laghari, and Rahul Yadav. "A Generative Adversarial Network Model Based on Intelligent Data Analytics for Music Emotion Recognition under IoT." Mobile Information Systems 2021 (2021).
Author Response
Dear expert teacher,
Hello!
Thank you very much for your care of our research group, and also for your very strict and professional review and guidance of manuscript Electronics-1813180.
Please see the attachment.
I wish you good health, smooth work and a happy family!
Kind regards,
All authors of manuscript Electronics-1813180.

Reviewer 2 Report
This paper proposes a solution for an edge-cloud collaborative system task offloading overhead model based on multi-vehicle users and multi-MEC servers. To minimize the delay and energy consumption of system execution tasks the total cost minimization problem is transformed into an optimization problem based on deep reinforcement learning (DRL). This method is deployed at the edge service layer to make fast offloading decisions for tasks generated by vehicle users. The simulation results show that the performance is better than the Deep Q-network (DQN) method and the Actor-Critic method regarding reward value and convergence. The authors claim that the proposed method has better performance in reducing the total computational cost, computing delay and energy consumption. Although the manuscript has merit, the authors should address the following comments for further improvement.
a. The title should remove the word DDPG. Better use “Dynamic offloading method for mobile edge computing of Internet of Vehicles based on multi-vehicle users and multi-MEC servers”
b. The abstract should be re-written. Need to highlight the motivation, problems and proposed solution and outcomes instead of simply jumping what the authors have proposed.
c. The contributions should be written more precise way. It seems broad and general.
d. What is the justification of using deep reinforcement learning for offloading decisions?
e. In related work section some references are missing. Plz see Reference [Error! Reference source not found. At the end of the related work section the authors should summarize the problems of the existing works and what new in the proposed method.
f. Need to discuss about Deep Deterministic Policy (DDPG). What exactly means?
g. Complexity of Algorithm 1 should be discussed.
h. Simulation environment should be discussed
i. What is the limitations of the work?
Author Response
Dear expert teacher,
Hello!
Thank you very much for your care of our research group, and also for your very strict and professional review and guidance of manuscript Electronics-1813180.
Please see the attachment
I wish you good health, smooth work and a happy family!
Kind regards,
All authors of manuscript Electronics-1813180.

Round 2
Reviewer 2 Report
Thank you so much for revising the manuscript and addressing the comments. The paper is now in good shape. Good luck!!!